# A Review on Topology Optimization Strategies for Additively Manufactured Continuous Fiber-Reinforced Composite Structures

Yogesh Gandhi * and Giangiacomo Minak 

Department of Industrial Engineering, Alma Mater Studiorum, Universitá di Bologna, Via Fontanelle 40, 47121 Forlì, Italy
* Correspondence: yogesh.gandhi@unibo.it

**Abstract:** Topology Optimization (TO) recently gained importance due to the development of Additive Manufacturing (AM) processes that produce components with good mechanical properties. Among all additive manufacturing technologies, continuous fiber fused filament fabrication (CF4) can fabricate high-performance composites compared to those manufactured with conventional technologies. In addition, AM provides the excellent advantage of a high degree of reconfigurability, which is in high demand to support the immediate short-term manufacturing chain in medical, transportation, and other industrial applications. CF4 enables the fabrication of continuous fiber-reinforced composite (FRC) materials structures. Moreover, it allows us to integrate topology optimization strategies to design realizable CFRC structures for a given performance. Various TO strategies for attaining lightweight and high-performance designs have been proposed in the literature, exploiting AM's design freedom. Therefore, this paper attempts to address works related to strategies employed to obtain optimal FRC structures. This paper intends to review and compare existing methods, analyze their similarities and dissimilarities, and discuss challenges and future trends in this field.

**Keywords:** topology optimization; continuous fiber-reinforced composites; additive manufacturing; structure optimization; fused filament fabrication



## 1. Introduction

The cost-effective, commercially available additive manufacturing (AM) technologies, or 3D printing (3DP), eliminate many limitations that previously plagued the manufacturing of highly tailored structural performance for multi-functional [1] and multi-physics [2] applications. Moreover, AM offers unique capabilities to realize that next-generation lightweight structures have brought great application potentials to several major industries such as the aerospace [3,4], automotive [5], and medical [6] sectors. First, AM techniques have the unique ability to fabricate highly complex shapes without a substantial increase in fabrication costs; in addition, the benefit of reducing manufacturing preparation time renders these technologies viable for large-scale industries. Moreover, it offers lattice structures, which are lightweight designs compared to solid-filled parts. Thus, AM offers the diversification of design to answer the requirements of multifunctional materials, such as weight reduction [7] and the ability to dissipate energy [8], heat [9], and vibrations [10].

AM can be classified based on the mechanisms and scopes of different assembling processes, such as material extrusion, vat polymerization, powder bed fusion, material jetting, binder jetting, and sheet lamination. However, material extrusion, referred to as fused filament fabrication (FFF), has the advantage over others because of its low costs and short production cycles. FFF-printed polymer parts frequently consist of carbon nanotubes and short fibers to upgrade their mechanical performance. Still, it cannot outperform [11,12] the mechanical strength offered by continuous fiber-reinforced composite laminate manufactured using conventional manufacturing tools. Hence, the shortcomings

of FFF-printed polymer composites support the development of continuous fiber filament fabrication (CF4). CF4 provides a unique opportunity to reduce distortion warping in parts and support structures during printing, and fiber tension prevents nozzle clogging, a constant challenge with polymer AM techniques. Additionally, controlling the anisotropic properties of FRCs can effectively distribute the loads throughout the laminate to maximize the strength and stiffness of the fabricated structures.

CF4 can accommodate a coaxial or dual extruder system in which the impregnation of fiber filaments occurs during printing. Thus, the categorization of CF4 can be based on impregnated fiber filaments [13], i.e., out-of-nozzle impregnation, in-nozzle impregnation, and semi-impregnated FRC filaments. Commonly, thermoplastic filaments used in this process are amorphous, with acrylonitrile butadiene styrene (ABS), poly-lactic acid (PLA), and PEEK (poly-ether-ether-ketone) being the most common, and the continuous fiber could be carbon, glass, natural fibers, etc. [14]. Several studies [15–17] compare the CF4 printed part to the same structure manufactured by traditional processes, which exhibits a higher mechanical performance than the CF4 part. These studies can address the limitation of existing CF4, i.e., the inability to ensure strong interlamellar adhesion between adjacent layers in the build direction. Therefore, the delamination tendency is higher due to poor inter-layer adhesion. Furthermore, void formation is intrinsic for several reasons: the heterogeneous diameters of the filament, uneven matrix distribution, poor filament impregnation, and fiber-rich regions. In addition, the layer-by-layer process and the printed bead's shape cause imperfect overlapping of the beads and void formation between the adjacent beads and layers. These voids act as structural defects and are responsible for prospective structural failure. Lastly, a critical review of CF4, including its mechanisms, investigations of CF4 materials, and process parameters, are detailed here [18].

Despite its limitations, CF4 allows the fabrication of FRC material with continuous spatial in-plane variations in the fiber angle and volume fraction, thus expanding the design space compared to that for variable [19] and constant stiffness laminate [20]. Moreover, CF4 technology can achieve out-of-plane variations in fiber angle due to the fiber-reinforced composite's self-supporting characteristics. Numerous studies have shown that fiber orientation optimization can significantly tailor structural performance, such as stress concentration [21], stiffness [22], buckling load [23], and natural frequency [24]. Therefore, the design of FRC structures requires optimization methods that reflect the design freedom offered by CF4 technologies, as well as its constraints, to thoroughly exploit the anisotropic properties of FRC materials [25]. These advantages to enhancing printed parts' overall functional performance, in contrast to the components' geometric-driven or/and cost-driven manufacturing, brought about a concept of performance-driven manufacturing named Design for Additive Manufacturing (DfAM) [26].

Topology optimization (TO), one of the DfAM methods, is an iterative design tool used to optimize a quantifiable objective while sustaining loads, constraints, and boundary conditions. TO is frequently adopted to design structurally sound parts and has subsequently surpassed design tools, such as shape and size optimization, in isolation. The seminal work of Bendsøe and Kikuchi [27] introduced the concept of TO for the homogenization method; since then, TO has developed rapidly. TO approaches can be summarized as follows: the homogenization method [27], the Solid Isotropic Material with Penalization (SIMP) method [28,29], the level set method [30,31], the Evolutionary Structural Optimization (ESO) method [32], and the Phase Field [33]. The details of these approaches are discussed in the review papers [34–36], and some emerging TO methods for smooth boundary representation include the 'Metamorphic Development Method' (MDM) [37] and the 'Moving Morphable Method' (MMM) [38]. The general architecture of TO starts with the definition of maximizing or minimizing a single or multi-target objective function to fulfill a set of constraints such as volume, displacement, or frequency [39]. Then, as part of an iterative process, design variables, finite element analysis, sensitivity analysis, regularization, and optimization steps are repeated until convergence is achieved [40].

The topology optimization concept applied to FRC structures enables optimizing material distribution, the orientation of fiber paths, material volume fractions, and even FRC material. In addition, for laminates, optimized stacking sequences and thickness are also considered. These attributes for the FRC structures are either optimized simultaneously or sequentially. Or sometimes, optimized FRC structures are attained, for simplicity, by selecting only one or few of the above attributes among the many variables that define FRC structure—given numerous parameters to optimize and extensive research on optimizing composites structures over the last decades [41,42]. This review focuses on parameterization schemes to consider material anisotropy in topology optimization. Thus, the present work does not aim to perform a comprehensive study on topology optimization, as extensive reviews on topology optimization approaches [35,43–45], their caliber to utilize them in various applications, and their realizability for additive manufacturing technologies [26,46] already exist. Therefore, we review the critical works for developing topology optimization methods for additively manufacturable fiber-reinforced composite structures or elaborate procedures that enable material anisotropy in available topology optimization approaches. Furthermore, several papers extend the suggested methodologies to study multi-physics and multi-objective measures or adapt them for numerical improvement or specific applications; we only provide essential references for brevity.

## 2. Topology Optimization for Continuum Structures

Generally, structural optimization can be categorized into three types: size, shape, and topology. Size optimization focuses on finding an optimal structural design by altering the size parameters of a structure or a component, such as the cross-sectional area of a truss bar, the thickness of a plane sheet, etc. On the other hand, shape optimization always works in a subset of allowable shapes of a structure with a fixed topology. It intends to optimize the structure's performance by changing the shape of its boundary.

Topology optimization is a numerical optimization technique to perform material distribution over design space subjected to boundary and loading conditions. TO problems are defined with a given set of performance criteria, constraints, and bounds on the design variables, which are fundamental quantities that are unknown and are optimized for the defined nested optimization problem. TO uses finite element analysis to evaluate the design performance, and the design is optimized using either gradient-based mathematical programming techniques or non-gradient-based algorithms. The method builds on repeated analysis and update steps, mostly guided by the gradient computation.

An objective function $\Phi$ represents the quantity being minimized or maximized to maximize the system's performance—the characteristics function $\chi_\omega$ associated with $\omega$ parameterizes the admissible topology $\mathcal{O}$ in the design domain, $\Omega \subseteq \mathbb{R}^d$, for the boundary value and optimization problem. The constraints $G_i$ are prescribed on the admissible topologies, thus making the problem well-posed. The design domain is an extended domain containing all the topologies, i.e., $\omega \subseteq \Omega \ \forall \ \omega \in \mathcal{O}$, and facilitates the description of the governing boundary value problem. The general optimization problem, then, can be written as:

$$
\begin{aligned}
\min_{\chi_\omega} &: \Phi(\chi_\omega, \mathbf{U}) \\
&:= \int_\Omega f(\chi_\omega, \mathbf{U}) d\mathbf{x} \\
\text{s.t.} &: G_0(\chi_\omega) = \int_\Omega \chi_\omega(\mathbf{x}) d\mathbf{x} - |\Omega_{\bar{d}}| \leq 0, \\
&: G_i(\chi_\omega, \mathbf{U}) \leq G_i^*, \quad i = 1, \ldots, K \\
&: \chi_\omega(\mathbf{x}) = \begin{cases} 0 \text{ or } 1 & \forall \mathbf{x} \in \Omega \end{cases}
\end{aligned}
\tag{1}
$$

In general, the objective function and constraints depend on the material distribution $\chi_\omega$ and the state variable $\mathbf{U}(\chi_\omega)$. Moreover, the nested approach considers the displacement functions' $\mathbf{U}(\chi_\omega)$ implicit dependency in the equilibrium equations, which are assumed to

fulfill each optimization step. Thus, to complete the discussion, $\mathbf{U}(\chi_\omega) \in \mathcal{V}$ satisfies the variational problem of elasticity:

$$\mathbf{U}(\chi_\omega) = \inf_{\mathbf{U}} \Pi(\mathbf{U}), \text{ such that}$$

$$\Pi(\mathbf{U}) = \int_\Omega \chi_\omega W_0(\mathbf{U}, \mathbf{x})\mathrm{d}\mathbf{x} - \int_{\tilde{\Gamma}_N} \boldsymbol{\tau} \cdot \mathbf{U}\mathrm{dS}, \tag{2}$$

where $\mathcal{V} = \left\{ \mathbf{U} \in H^1(\Omega, \mathbb{R}^2) : \mathbf{u}|_{\Gamma_D} = \mathbf{0} \right\}$ is the space of admissible displacements independent of $\omega$; $\Pi_{\Omega_d}(\mathbf{U})$ is the total potential energy of the system; $W_0(\mathbf{u}, \mathbf{x})$ is the strain energy density of the solid material present in $\Omega_d$; $\Gamma_D$ and $\Gamma_N$ form a partition of $\partial\Omega$; and $\boldsymbol{\tau}$ is the non-zero traction applied on $\tilde{\Gamma}_N \subseteq \Gamma_N$, as shown in Figure 1. It is well-known that optimal solutions to problem (1) are not guaranteed due to the lack of closeness of the set of feasible designs. Thus, we assume that there are design or manufacturing constraints imposed on $\mathcal{O}$, which makes the problem well-posed—commonly known as the restriction settings—in contrast to the relaxation strategy, as detailed in the review paper [47].

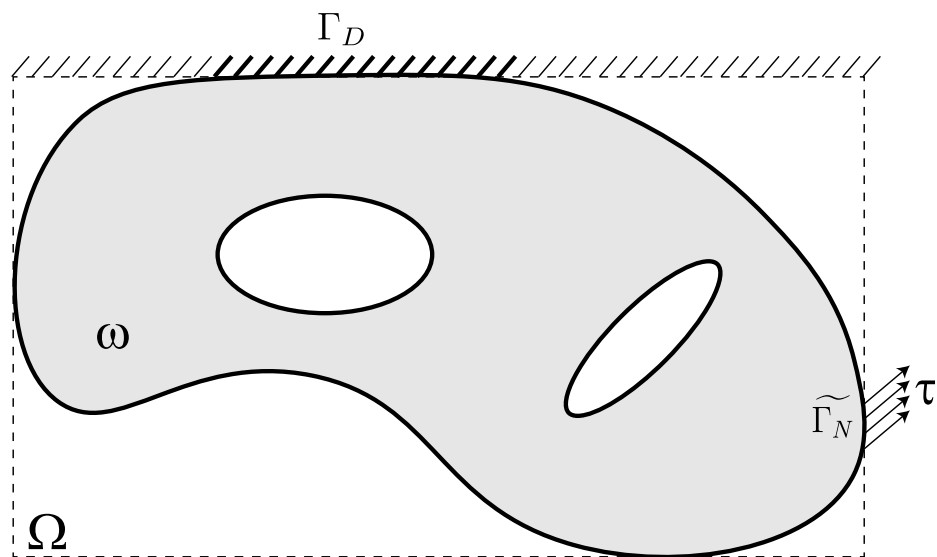

**Figure 1.** Extended design domain and boundary conditions for the state equation (adapted from [48]).

In the discretized design domain $\Omega_h$, parameterizing the admissible topologies via characteristic functions leads to an integer programming problem, as expressed later in Section 2.2. Moreover, such discrete settings are intractable for a large system; thus, the optimization variables in the general problem are primarily due to the continuous parameterization of the topology. For example, the characteristic function in the description of the state function can be replaced by the density function, $\rho$, that takes continuous values $[0, 1]$ or by the Heaviside function, $H(\phi)$, where an implicit function, $\phi$, belongs to the bounded interval $[-\alpha, \alpha]$ for $\alpha > 0$, defined in (3).

### 2.1. Shape-Based Topology Optimization

The general problem (1) can be tackled using Lagrangian approaches, such as the non-parametric shape optimization methods, where the nodal positions in the FE model represent geometry and are updated in the optimization process. The classical shape optimization exhibits a continuous mapping with a continuous inverse function between two topological spaces, i.e., homeomorphism exists. Therefore, it must be combined with a criterion to generate new holes, e.g., the bubble method [49], topological derivatives [50,51], etc., for topological changes to solve the general problem. However, the approach is rather challenging, partly because perturbing the design variables may adversely affect boundary

changes and partly due to re-meshing and adaptive meshing to track moving boundaries and interfaces. As reviewed here [52], readers can find several shape optimization techniques and their current developments. On the other hand, the recent advent of utilizing isogeometric analysis for shape and topology optimizations provides several advantages, such as a seamless integration between analysis and design, exact geometric representation, and non-parameterized structural boundaries. Interested readers can find several attributes of isogeometric shape and topology optimization in [53,54].

Generally, the Lagrangian formulation adopts alternative non-parametric techniques based on a free-form implicit design representation with level-set methods. The LSMs are a variant of shape optimization methods that operate with well-defined boundaries but are allowed to move for forming, removing, and merging void regions, which define the actual topological design. The LSMs define a level set function (LSF) with a higher dimension to represent the structure. The zero-level set, $\phi$, describes the material interface $\partial\Omega$; a LSF with negative values is applied to define the voids domain. The LSF describes the material domain, $\Omega_d$, in the design space, $\Omega$, with positive values as follows:

$$
\begin{cases}
\phi(\mathbf{x}, \tau) > 0 \Leftrightarrow \mathbf{x} \in \Omega \text{ (material)} \\
\phi(\mathbf{x}, \tau) = 0 \Leftrightarrow \mathbf{x} \in \Gamma \text{ (interface)} \\
\phi(\mathbf{x}, \tau) < 0 \Leftrightarrow \mathbf{x} \in (\Omega_d \backslash \Omega)( \text{ void })
\end{cases}
\tag{3}
$$

where $\tau$ denotes a pseudo time that represents the iteration in the optimization process. Hence, the evolution of LSF advances the structure's shape and possibly its topology in the material domain. Evolving the LSF in the optimization process is mainly governed via the solution of the Hamilton–Jacobi equation, which is first-order and models convection only:

$$
\frac{\partial \phi(\mathbf{x}, t)}{\partial t} - v_n \|\nabla \phi\| = 0, \quad \phi(\mathbf{x}, 0) = \phi_0(\mathbf{x})
\tag{4}
$$

where $v_n = \mathbf{v} \cdot \mathbf{n}$, such that the normal to the zero-level contour is related to the gradient of the LSF by $\mathbf{n} = \frac{-\nabla \phi}{\|\nabla \phi\|}$. For a more mathematical discussion, we refer the reader to the works of Burger and Osher [55–57].

A general form of level-set-based topology optimization, together with (2), in which the Heaviside function, $H(\phi)$, reflects the characteristic function, can be written as follows:

$$
\begin{aligned}
\min_{\phi} : &\ \Phi(\mathbf{U}, \phi) \\
:= &\ \int_{\Omega} f(\mathbf{U}(\phi)) H(\phi) d\mathbf{x} \\
\text{s.t.} : &\ G_0(\phi) = \int_{\Omega} H(\phi) d\mathbf{x} - |\Omega_d| \leq 0, \\
: &\ G_i(\phi, \mathbf{U}(\phi)) \leq G_i^*, \quad i = 1, \dots, Q \\
: &\ H(\phi) = \begin{cases} 0, & \phi(\mathbf{x}, \tau) < 0 \\ 1, & \phi(\mathbf{x}, \tau) \geq 0 \end{cases}
\end{aligned}
\tag{5}
$$

LSTO, as classified above, operates on the boundaries instead of local density, which is the zero-level set of a scalar function (LSF), and topological changes are based on the LSF's evolution. Moreover, several LSMs have emerged, classified based on LSF parameterization, mapping the level-set-based geometry onto the mechanical model, and the strategy for solving the optimization problem, as detailed in the review paper [35]. The identification of the material interfaces, the structural response's accuracy, and the optimization process's convergence are influenced by the structural boundaries in the discretized mechanical model. The structural topology is mapped using conforming mapping, immersed boundary techniques, or density-based mapping. The first two approaches generally provide a crisp representation of the mechanical model's boundaries and are considered to solve variational problems (2) to perform general topology optimization (5). On the other, density-based

mapping replaces the Heaviside function with density distribution $\rho(\phi)$ to approximate (5) as $\int_\Omega f H(\phi) dV \approx \int_\Omega f \rho(\phi) dV$, where the density distribution can map the LSF point-wise or element-wise.

Finally, we rest our synopsis on the LSTO on the following grounds. First, most of the works reviewed in the articles follow density-based TO (as discussed in Section 2.2) to optimize anisotropic materials' orientation and distribution; moreover, DTO has also been applied to several other applications in academia and industrial sectors. Secondly, LSTO often adopts Eulerian mesh with ersatz material (point-wise density distribution) and a DTO formulation because of the DTO's simplicity and ease of implementation. Still, the crisp boundary description is maintained throughout the optimization process, allowing shape sensitivity analysis and design updates by solving the HJ equation, i.e., different from DTO.

Notably, the above deduction is not biased toward following a particular TO approach as no comparative studies are performed on their methodologies, numerical efficiency, numerical verification of the attained optimized design, or their realizability and applicability. Hence, the readers are referred to citations marked earlier for different TO approaches.

### 2.2. Density-Based Topology Optimization

Earlier in this section, the general problem was formulated based on the boundary-following mesh or Lagrangian model. In contrast, the Eulerian model (fixed mesh) maps the topology through a density function, $\rho$, on the discretized design domain. Finally, this density distribution, $\boldsymbol{\rho}$, is fed to the optimization process as design variables. Such reformulation is called density-based topology optimization. Compared to the shape-based TO, the nodes of the structures' boundaries are the optimization variables, and LSTO considers the parameterized LSFs to be the design variables for the optimization process. Thus, the structure's boundaries are at least retained in the optimization process, even though it has blurred out the discretized setting of the continuum design space when utilizing density-based mapping. However, when elements or nodes of the mesh are optimized, such as sizing variables, the sense of the exact illustration of the structure's boundaries is physically lost. Mathematically, it takes the following form:

$$
\begin{aligned}
\min_{\boldsymbol{\rho}} &: \Phi(\mathbf{U}, \boldsymbol{\rho}) \\
&:= \sum_i \int_{\Omega_i} f(\rho_i, \mathbf{U}(\rho_i)) d\mathbf{x} \\
\text{s.t.} &: G_0(\boldsymbol{\rho}) = \sum_i \mathbf{v}_i \rho_i - |\Omega_d| \leq 0, \\
&: G_j(\boldsymbol{\rho}, \mathbf{U}(\boldsymbol{\rho})) \leq G_j^*, \quad j = 1, \ldots, Q \\
&: \rho = \begin{cases} \epsilon, & (\text{ void }) \\ 1, & (\text{ solid }) \end{cases}, i = 1, \ldots, N
\end{aligned}
\tag{6}
$$

The density distribution, $\boldsymbol{\rho}$, denotes the design variable vector of length N. It is defined such that $\rho = 1$ if $\mathbf{x} \in \omega$ and $\rho = 0$ otherwise. For regions where $\rho = 0$, solutions to the boundary value problem are not guaranteed, as the energy's bilinear form is not coercive. Thus, the density function is defined as $\epsilon + (1 - \epsilon)\boldsymbol{\rho}$, in which $\epsilon \ll 1$ is the ersatz parameter:

$$
\begin{aligned}
&\mathbf{U} = \inf_{\mathbf{U}} \Pi(\boldsymbol{\rho}, \mathbf{U}), \text{ s.t.} \\
&\Pi(\boldsymbol{\rho}, \mathbf{U}) = \int_\Omega [\epsilon + (1 - \epsilon)\boldsymbol{\rho}] W_0(\mathbf{U}, \mathbf{x}) d\mathbf{x} - \int_{\bar{\Gamma}_N} \boldsymbol{\tau} \cdot \mathbf{U} dS
\end{aligned}
\tag{7}
$$

Thus, the above formulation is a binary problem representing the void and solid regions of the structure, hence named discrete density-based topology optimization (DDTO). The well-known DDTO method is the Bi-directional Evolutionary Structural Optimization (BESO), as it presently defines and adopts techniques commonly used in continuous

DTO approaches. Recently, Sivapuram et al. [58] combined the features of BESO and the sequential integer linear programming for discrete topology optimization. Interested readers can find comprehensive reviews on the BESO methods in [44,59]. Another outlook on approaching the discretizing problem is using a genetic algorithm [60] that can find a "global minimum" and allow the handling of a discrete variable, but this always comes at a higher computational cost. Furthermore, Sigmund [61] questions the usefulness of non-gradient approaches in TO. With that, we concluded the discussion on DDTO as, in practice, it is recommended, when formulating the TO problem, to assume the continuous density field together with (7), as portrayed below:

$$
\begin{aligned}
\min_{\boldsymbol{\rho}} : &\ \Phi(\mathbf{U}, \boldsymbol{\rho}) \\
:= &\ \sum_i \int_{\Omega_i} f(\rho_i, \mathbf{U}(\rho_i)) d\mathbf{x} \\
\text{s.t.} : &\ G_0(\boldsymbol{\rho}) = \sum_i \mathrm{v}_i \rho_i - |\Omega_d| \leq 0, \\
: &\ G_j(\boldsymbol{\rho}, \mathbf{U}(\boldsymbol{\rho})) \leq G_j^*, \quad j = 1, \ldots, Q \\
: &\ 0 \leq \rho_{min} \leq \rho \leq 1, \quad i = 1, \ldots, N
\end{aligned}
\tag{8}
$$

The above Formulation (8) is a broadly received idea in the TO of continuum structures that utilizes continuous density design variables instead of binary density variables, thus enabling the use of gradient-based information. The density function, $\rho$, takes values in $[0,1]$ and replaces the characteristic function in the description of the state Equation (9) and the objective and constraint functions. In addition, to attain binary design, the density function interpolates the material properties through the material interpolation function as given in the state equation, where $p > 1$ is a penalization exponent [28,29]:

$$
\begin{aligned}
\mathbf{U} = &\ \inf_{\mathbf{U}} \Pi(\boldsymbol{\rho}, \mathbf{U}), \ \text{s.t.} \\
\Pi(\boldsymbol{\rho}, \mathbf{U}) = &\ \int_{\Omega} [\epsilon + (1-\epsilon)\boldsymbol{\rho}^p] W_0(\mathbf{U}, \mathbf{x}) d\mathbf{x} - \int_{\tilde{\Gamma}_N} \boldsymbol{\tau} \cdot \mathbf{U} dS.
\end{aligned}
\tag{9}
$$

Still, the optimal solution to this problem, in general, is not guaranteed because there is a lack of closedness in the set of feasible designs space, i.e., generating even more holes will decrease the objective function. In addition, in the discretized space, numerical instabilities arise, including checkerboarding and mesh dependency. Checkerboarding refers to forming patches of alternating solid–void elements, whereas mesh dependency causes different topologies from similar design domains of different discretization sizes. Therefore, restrictions are imposed on the admissible density function in practice to prevent the rapid oscillation of the density distribution, as suggested in the papers [47,62]—in contrast to the relaxation settings that accommodate generalized shapes due to severe oscillation of the density distribution. The concept refers to the homogenization approach to topology optimization [27]. Notably, regularization strategies are imposed similarly on the variation in the LSF for well-posedness.

### 2.3. Gradient-Based Update Schemes for Topology Optimization

Reasonably arranging the fiber orientation is critical to effectively handling an anisotropic material, which is vital for designing next-generation lightweight composite structures. Frequently, fiber orientation optimization creates a difficulty associated with local optima and discontinuous functions. Thus, to address this, gradient-free algorithms [63] are more qualified because of their global searching ability [64,65]. Furthermore, by allowing differentiable functions, mixed design variables and discrete space introduce a relaxed formulation that has the advantage of obtaining fewer local optima. However, the inefficiency of most gradient-free algorithms requires numerous function evaluations, which is impractical for expensive finite element simulations. Hence, the adoption of gradient-based algorithms, i.e., Optimality Criteria Method (OCM), Method of

Moving Asymptotes (MMA) [66], and Sequential Linear Programming (SLP) [67], becomes a reasonable choice for the TO problems.

The OCM is derived using the Lagrange function, which is composed of objective and constraint functions that satisfy the Karush–Kuhn–Tucker (KKT) condition for an optimal solution. The OCM procedure has double loops, where the inner loop updates the design variable, and the outer loop updates the Lagrange multiplier based on the KKT condition. However, the method cannot handle multiple constraints because the coupling of the Lagrange multiplier and the design variables requires solving a nonlinear equation. Therefore, Shen et al. [68] questioned the lack of understanding about the orientation optimization algorithm to handle arbitrary constraints and loads in the OCM. A step length scheme for orientation optimization is advised to achieve global descent by normalizing the gradient vector and introducing a parameter to control the magnitude of material orientation in each iteration. However, the verification lacks the effect of adding constraints in the orientation optimization problem on the update scheme, a critical factor for the OCM. Thus, a more generalized OCM for the topology optimization of an anisotropic material is demanded from scalability and multiloading situations. Recently, Kim et al. [69] interpreted the work of Patnaik et al. [70] on parametric optimization and proposed a generalized optimality criteria method for topology optimization problems. The approach eliminates the compulsion to satisfy the constraints during every optimization iteration but should be met upon convergence.

On the other hand, SLP and MMA are general-purpose optimization strategies supporting various multi-objective and multi-constraint nonlinear problems (NLP) in engineering. In these first-order methods, the gradient information about a design point approximates the constraint and objective functions. In particular, for MMA, a hybrid form of the linear and reciprocal approximation [71] has the advantage of being convex, which introduced the term convex linearization (CONLIN) [72] for approximating the optimization problem. Svanberg introduces a convex approximation variation that stabilizes and speeds up the convergence of process optimization by controlling moving asymptotes while the approximation remains convex and first-order. Furthermore, because the subproblem is separable and convex, a dual approach or a primal-dual interior-point method can efficiently solve the NLP. However, the reciprocal approximation in MMA might eliminate the linearity of approximation [73].

## 3. Parameterization Schemes for Fiber Orientation

The parameterization scheme implements a numerical description of fiber orientation patterns and defines variables for optimization. It should ensure the spatial continuity of fiber angles so that the CF4 technology can produce the structure. It should also provide enough design freedom so that the optimization algorithm can consider more candidate designs.

For mathematical completeness, the general density-based template for the optimization problem is presented to find the optimal distribution of structural topology, fiber layout, and fiber orientation in functionally graded anisotropic composite structures. In Figure 2, the density distribution accommodates fiber material only; however, the addition of materials is considered through separate density functions in the optimization framework, for example, when optimizing variable fiber fractions or functionally graded anisotropic composites, as depicted in the Equation (10). Thus, as detailed in this section, the template accommodates several parameterization schemes used in the literature to optimize fiber-reinforced composite structures.

$$\min_{z} : \Phi(\mathbf{U}, z)$$

$$:= \sum_{i} \int_{\Omega_i} f(z_i, \mathbf{U}(z_i)) d\mathbf{x}$$

$$\text{s.t. } : z := \left[ \boldsymbol{\rho}_m, \boldsymbol{\rho}_f(\theta) \right]$$

$$: \underline{z}_i \leq z_i \leq \bar{z}_i, \forall z_i \in z, \quad i = 1, \dots, N \tag{10}$$

$$: G_{m0}(\boldsymbol{\rho}_m) = \sum_{i} v_i \rho_m^i - V_m \leq 0,$$

$$: G_{f0}(\boldsymbol{\rho}_f) = \sum_{i} v_i \rho_f^i - V_f \leq 0,$$

$$: G_j(z, \mathbf{U}(z)) \leq G_j^*, \quad j = 1, \dots, Q$$

$$: K(z)\mathbf{U} = F$$

The vector $z$ contains all the design variables, i.e., isotropic material (matrix) density, fiber material density, and orientation variables. Each design variable is bound between the values $\underline{z}_i$ and $\bar{z}_i$. The total volume of the matrix material, $G_{m0}$, is calculated from the density distribution, $\boldsymbol{\rho}_m$, determined by the design variable (or density function) $\boldsymbol{\rho}_m$. Similarly, the fiber material volume is determined through the fiber density function $\rho(\theta)_f$. The system of linear equations is composed of the stiffness matrix, $K$, and the force vector, $F$, which is derived from the state equation by finite element formulation.

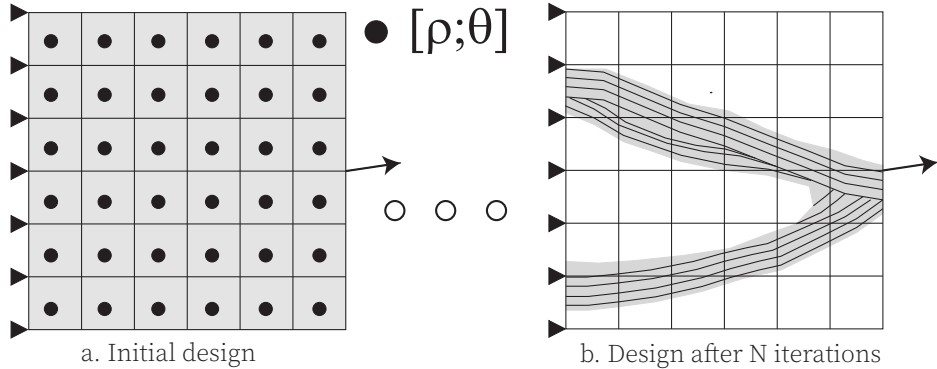

a. Initial design          b. Design after N iterations

**Figure 2.** Illustration of DTO considering anisotropic material via fiber orientation parameterization.

### 3.1. Continuous Parameterization

The continuous parameterization of fiber orientation (CFO) design uses the angle itself as the design variable [74,75]. The design variable is the continuous and independent parameter that provides flexibility in changing the orientation across the design points, relaxing orientation design space, as shown in Figure 3. The rotation stiffness tensor, $\overline{\mathbf{C}}(\theta)$, is derived from the base anisotropic stiffness tensor, $\mathbf{C}$, using rotation tensor $\mathbf{T}(\theta)$, where $\theta$ coincides with the direction of the fiber, and $c$ and $s$ stands for $\cos\theta$ and $\sin\theta$:

$$\overline{\mathbf{C}}(\theta) = \mathbf{T}^{-1}(\theta) \cdot \mathbf{C} \cdot \mathbf{T}'(\theta) = \begin{bmatrix} \bar{C}_{11} & \bar{C}_{12} & \bar{C}_{16} \\ \bar{C}_{12} & \bar{C}_{22} & \bar{C}_{26} \\ \bar{C}_{16} & \bar{C}_{26} & \bar{C}_{66} \end{bmatrix} \tag{11}$$

$$\mathbf{T}(\theta) = \begin{bmatrix} c^2 & s^2 & 2cs \\ s^2 & c^2 & -2cs \\ -cs & cs & c^2 - s^2 \end{bmatrix}, \quad \mathbf{T}'(\theta) = \begin{bmatrix} c^2 & s^2 & cs \\ s^2 & c^2 & -cs \\ -2cs & 2cs & c^2 - s^2 \end{bmatrix} \tag{12}$$

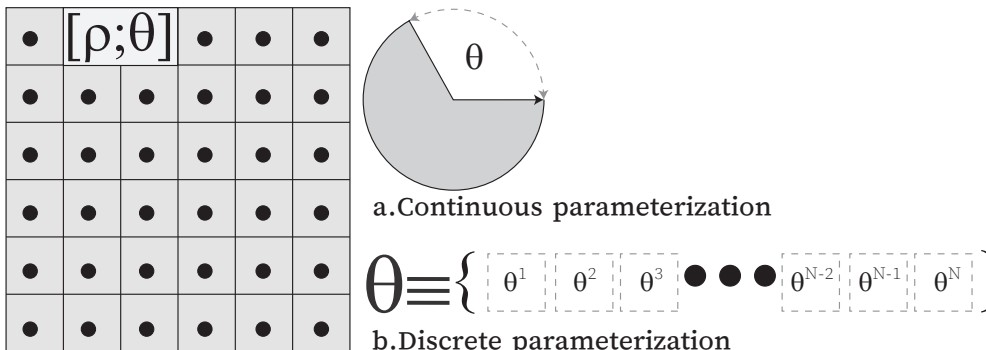

**Figure 3.** Finite elements are considered design variables in DTO: (**a**) Continuous material orientation; (**b**) Discrete material orientation.

Paramaterized fiber orientation, $\theta$, in the optimization process continuously spanned in the angle range $\left[\frac{-\pi}{2}, \frac{\pi}{2}\right]$. Thus, handling a continuous fiber orientation design presents difficulties due to a fourth-order transform tensor that rotates to a given angle composed of multivalued sine and cosine functions, resulting in a non-convex optimization problem. Furthermore, optimizing the fiber orientation is susceptible to the initial fiber configuration, thus causing difficulties in obtaining the optimized solution. As illustrated in [76], suboptimal solutions are the persistent outcome of a continuous fiber orientation design problem. One brute-force way to avoid it is by further relaxing the design space. For instance, free material optimization (FMO) [77,78] parameterizes each stiffness tensor element independently as the design variable. This secures the scheme from the complexity of the orientation design variable of the design space. However, as compensation, pointwise nonlinear constraints ensure the positive semi-definitiveness of the obtained stiffness tensor and link it to the feasible physical design, making this approach challenging. Nomura et al. [79] formulated an orientation design variable as a tensor field to simplify the first tensor invariant constraint and remove nonlinear constraints successfully introduced due to the second tensor invariant. Still, as commented, the violation of these constraints is observed at the joint point of the structural members where the orientation shows the discontinuous distribution.

Early studies utilizing the analytically derived optimally criterion [80] for optimizing fiber orientation date back to the pioneering work of Pedersen on the strain-based method [81–83]. In that work, strain energy density was transformed into principal strain, and it was concluded that material orientation axes that lie along principal strain axes always give stationary energy density. However, Cheng [84] argued that the discussion is limited to a unit cell case where the orientation variable is separated from the design domain to obtain extreme strain energy. After that, a similar deduction using iterative optimality criteria [85,86] formulated the stress-based method [87] by exercising an invariant stress field for material orientation. Finally, Diaz and Bendsoe [88] extended the stress-based method to determine the optimal orientation optimization problem corresponding to multiple loads. Despite their similarity, the stress-based method produces a slightly stiffer structure than the strain method because strong couplings exist among the orientational variables when the strain field is used [84]. Conclusively, Gea and Luo [89] demonstrated that the fiber orientation coincides with the principal stress/strain fields for relatively weak shear and some strong shear types of anisotropic materials. Recently [90], a strain-based method framework has been utilized to optimize laminate topology and fiber orientation for various in-plane and out-of-plane loading conditions.

Furthermore, the CFO methods are highly dependent on the initial fiber configuration. As drawn out here [91], these approaches will fail for shear 'strong' type materials due to repeated global minimum solutions. Nevertheless, the shortcomings of these methods encouraged the formulation of the energy-based method introduced by Luo and Gea [92,93]. This method uses an inclusion cell to estimate the strain and stress fields' dependency on the fiber orientation by introducing an approximate energy factor. Yet, the

dependence of energy factors on the traction stress, material properties, and direction of the inclusion cell and its surroundings makes it challenging to formulate the framework for 3D and complex loading problems. Following the principles of the energy-based method, Yan et al. [94] proposed a hybrid stress–strain method by weighting the optimality condition of the mean compliance in the stress and strain form. Numerical examples demonstrate their method on weak and strong shear materials and extension to 3D problems. The assumption regarding the elemental strain and stress field invariant to the neighboring elemental orientation is considered; however, it may restrict the solution of 3D problems and result in a suboptimal solution.

An alternative is employing curvilinear parameterization schemes that define fiber paths as the graphs of analytical function, which guarantee the continuity of the fiber angle and have a small number of design variables [95–97]. Nevertheless, the restrictive design search space will limit the tailoring of the fiber path, thus deteriorating the stability of the optimization problem [19] and the quality of the optimized solution. In addition, the parameterization schemes can follow equidistant iso-contours of a level set function to represent curvilinear fiber paths [98,99], naturally ensuring fiber continuity and often being parallel to the neighboring fiber paths. Furthermore, the optimization result becomes highly dependent on the initial configuration, and local solutions often appear [100].

### 3.2. Discrete Parameterization

The counter scheme restricts the orientation design space through a discrete orientation optimization formulation to avoid multiple local optima issues. Initially, it was solved using a genetic algorithm at the cost of a computational burden [101–103]. Thus, Lund [76] relaxes the combinatorial problem to a continuous optimization problem. The discrete material candidates are chosen a priori through transforming the base anisotropic stiffness tensor for given fiber orientations. Finally, the effective anisotropy elasticity tensor is calculated as a convex combination of material candidates and satisfies the following conditions, as shown below:

$$
\mathbf{C}_{eff} = \sum_{i=1}^{n_c} w_i \overline{\mathbf{C}}_i, \quad
\begin{aligned}
& 0 \leqslant w_i \leq 1 \\
& \sum_{i=1}^{n_c} w_i = 1
\end{aligned}
\tag{13}
$$

Thus, the scheme shares some similarities with the multi-material optimization problem in [104,105]. The suggested scheme assigns weighting functions to different candidates and employs gradient-based optimization with a penalization coefficient, forcing the weighting functions to seek a binary design and fiber convergence, i.e., one discrete material at each design point. This method is known as Discrete Material optimization (DMO). DMO laid the foundation for shape function with penalization (SFP) [106] and bi-value coding parameterization (BCP) [107] to perform discrete fiber orientation optimization. Later, DMO was extended for laminated composite structures to determine the material distribution and thickness variation, known as discrete material and thickness optimization (DMTO) [108]. Some recent works [109–111] further improved the applicability of DMTO. A comparison of these methodologies using various numerical examples is contained in [111,112]. Another work proposed a self-penalization interpolation model for fiber orientation (SPIMFO) based on convergent Taylor series for sine and cosine functions to optimize composite hyperelastic material [113] and the dynamic design of laminated piezo-composite actuators [114].

DMO does not incorporate design problems for continuously varying orientation distributions. First, it is an imperative design consideration to circumvent stress constraints and degradation in the strength by order of magnitude compared to that for continuous fiber paths due to fiber discontinuity. Secondly, these methods fail to address the fiber convergence, even against the significant penalization factor; hence, their benefit relies on an optimization algorithm to circumvent impractical mixtures of fiber orientations. Third, the discrete parameterization schemes should further minimize the number of design candidates for efficient optimization.

### 3.3. Discrete-Continuous Parameterization

Utilizing continuous and discrete methodology benefits is another alternative to fiber orientation optimization. The key idea in the following approaches is to fill the gaps by acknowledging the beneficial characteristics of both strategies to improve computational efficiency, reduce local optima, and/or resolve fiber continuity and/or manufacturability issues. Therefore, an approach to reduce the risk of falling into local optimal without sacrificing the fiber's continuity can use both discrete and continuous parameterization, as suggested by Luo et al. [115]. Their work proposed a coarse-to-fine strategy, where the orientation design space is divided into discrete sub-intervals. After that, the CFO searches for an optimized solution in a sub-interval, where the sub-interval selection problem is solved using the DMO approach. However, no criterion is defined to determine the number of sub-intervals required in advance.

Nevertheless, the proposed strategy provides flexibility to integrate alternatives suggested for DMO and CFO approaches. Nomura et al. [116] studied the Cartesian system for orientation design variables to improve initial design dependency and local optima issues encountered in the continuous parameterization approach. The parameterization scheme was further extended to yield an optimized discrete orientation design for a given discrete orientation set in their work. Moreover, the characteristics representing the orientation design variables in the vectorial form consider the $2\pi$ ambiguity, which occurs due to the periodic nature of the orientation design variable. Introducing vectorial design variable as a point-wise quadratic inequality constraint yields more interpolated elasticity tensors than the single variable polar representation. However, the optimization algorithm treats continuous and discrete problems as two different problems; thus, investigating the coupled optimization framework might be an outlook for consideration. Kiyono et al. [112] proposed a parameterizing scheme that continues the computational approach suggested by Yin and Ananthasuresh [105]. Introducing a normal distribution function as a weighting function in their parameterizing strategy guarantees fiber convergence, low sensitivity to the initial fiber configuration, and continuity of the fiber orientation.

Xia and Shi [117] develop a continuous global function by applying the shepherd interpolation method at scattered design points to represent the fiber orientation throughout the design domain. The interpolation function's benefit is that it ensures fiber continuity while considering a reduced orientation design space in contrast with CFO. Unfortunately, it suffers from the initial configuration and ends at the sub-optimal solution. Another work of Xia [118] applied multilevel optimization for fiber orientation optimization and verified its efficiency against single-level optimization. Still, the optimization results in different fiber arrangements for different initial fiber orientations. As a result, the efficiency of the multilevel approach relies on the attained fiber orientation field at a coarse level since the optimization at the successive refined level starts from an initial design computed at its neighboring coarser level.

A more recent effort by Ding et al. [119] introduced a discrete-continuous framework taking inspiration from the available literature [112,115]. However, no comparative studies are considered, and solutions fall to the local minima when the discrete-continuous interval is greater than two. Finally, Qiu et al. [91] present an approach to optimizing material orientations using multiple print planes (Figure 4) and demonstrate their optimization framework extends the material orientation design space by attaining lower compliance value. Apart from the numerical verification, the results are manufactured using nylon filaments with chopped carbon fibers. Lastly, the choice of a discrete-continuous interval is illustrated through an example.

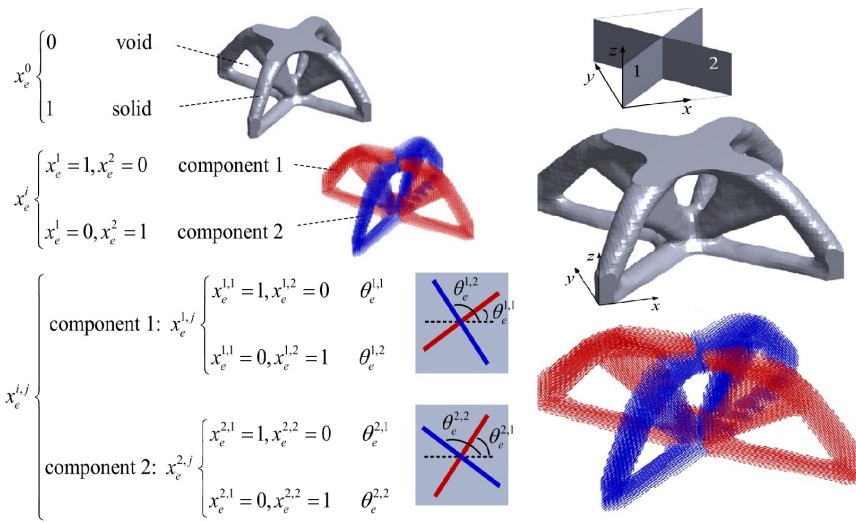

**Figure 4.** (**Left**) Schematic diagram represents design variables of an element and parameterizing continuous orientation variable in the discrete-continuous setting. (**Right**) Implementation of the multiple print plane to design MBB [91].

### 3.4. Feature-Based Parameterization

The parameterization schemes, as mentioned earlier, introduce low-level fiber material representations, such as pixel or voxel-based representations, thus representing the designs with variables proportional to the number of pixels or voxels in the design space. Moreover, these techniques render organic and free-form designs, which require sophisticated postprocessing to distinguish fiber paths for the use of CF4. Therefore, to avail manufacturable solutions with designs containing few variables, fiber material can be considered as a geometric feature with high-level parameters. High-level parameters refer to spatial dimensions associated with a feature's size, position, or orientation. Finally, feature-mapping techniques map these features onto a fixed mesh for analysis; an extensive review of feature-mapping methods by Wein et al. [45] details the components of feature-mapping techniques and discusses their implementation in structural optimization. Geometry projection [120] is an explicit feature-mapping technique extended to represent the design via cylindrical bars reinforced with continuous fibers [121] and performs the analysis using a fixed finite element mesh. The interpolation of the material properties at the junction of multiple bars made of an anisotropic material is penalized as a convex combination of the penalized effective densities for each component. Furthermore, it demonstrates that the method can easily integrate shape constraints on the structural form offered by CF4 as shown in Figure 5. Thus, this work introduces the groundwork for using the geometry projection method for fiber-orientation optimization design problems. However, the fiber is restricted to being unidirectional along the bars, further constraining the possibility of exploiting CF4's design freedom. In the same spirit, the method can be used to accurately control the structure's size through the explicit representation of features. For example, Sun et al. [122] proposed a trapezoidal component made of primary material and wound by the fiber layers. However, the framework results in small features size appearing in numerical problems. The Table 1 compares parametrization schemes, stating their allowed degree of freedom in the design space, advantages and disadvantages in finding an optimized design for continuous FRCs material, and applicability to CF4.

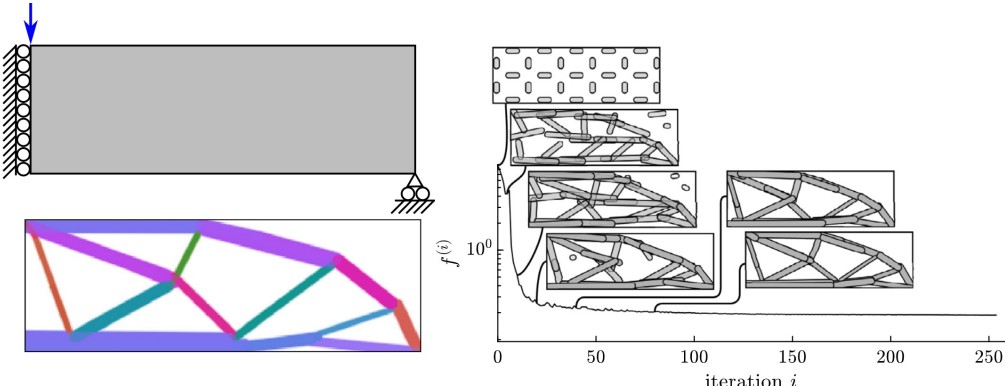

**Figure 5.** The example demonstrates the design of MBB using the geometry projection method (adapted from [121]). The top of the left column shows the MBB beam design region, support, and unit load, whereas the bottom depicts colored penalized element densities for the optimal MBB beam designs. Note that the color denotes the orientation of the rest of the changes. In the right column, iteration histories of an objective function indicate the attained compliance value $f^{(i)}$ at an iteration $i$.

**Table 1.** Various attributes and comparison of enumerated parameterization techniques.

| Parameterization | Design Freedom | Advantages | Applicability | Drawbacks |
|---|---|---|---|---|
| Continuous | Fully relaxed material orientation space. | Spatially varying fiber path both in 2D and 3D. | Adopted scheme for CF4 part design and verification. | Initial design dependency, significant variation in fiber angles, and $\pi$ ambiguity result in poor local minima. |
| Discrete | Most restrictive material orientation space. | Most effective gradient-based method for discrete settings [1], e.g., multi-phase TO. | Numerous studies on designing multi-layered composite laminates. | Several design variables, fiber convergence, and material discontinuity lead to ambiguous design. |
| Discrete-continuous | Continuous orientations are penalized for attaining assigned discrete directions. | General framework for both continuous and discrete settings. | Promising framework to withstand various FRC manufacturing units. | Only a few works are available, and an efficient optimization formulation is needed to tackle the general setting. |
| Feature-based | Most restrictive material distribution space [2]. | Least number of optimization variables, easy-to-control feature size, and ready-to-manufacture design. | Favorable for large-scale application and industrial manufacturing units due to its simple topology. | Topology is restricted [3] to feature shape, thus limiting CF4 capabilities |
| Material | Completely relaxed in material distribution and orientation space | Allows spatially varying fiber path and volume fraction, complex loading, integration of failure criteria, damage model, etc. | Fully exploit the capabilities of CF4 with a design that can follow the response of the actual part. | Validation of numerical framework is difficult because the attained topology is complex |

[1] Genetic algorithms are used in the discrete framework; [2] Mainly for explicit feature-mapping; [3] Implicit features can allow free-form topology at the cost of computing the distance function numerically.

### 3.5. Material Parameterization

Further relaxation in the feasible design space can introduce through material heterogeneity. For example, heterogeneous composite [123] materials consist of two or more materials and are engineered to vary the spatial composition and structure continuously. Thus, such variation pushes the envelope of material property space beyond natural limits. In addition, recent studies [124–126] have shown that CF4 is ready to manufacture FRC structures with continuous yet spatially varying fiber paths and fiber volume fractions. Thus, if properly optimized, variable FRC material properties may perform better than a fixed FRC material volume fraction. Therefore, a composite structure comprising heterogeneous FRC material distribution provides considerably larger design freedom to CF4. Accordingly, Li et al. [127] considered a SIMP-based sequential TO approach to design FRC structures by considering fiber and material fractions in a given design space. A sequential process begins with designing an isotropic-material matrix with voids, selecting fiber fractions, and optimally orienting the fibers. However, the sequential approach sacrifices the exploration of new topologies that might be optimal for variable FRC structures. In that spirit, Desai et al.'s [128] work investigated the simultaneous design of matrix topology, fiber material layout, and orientation using an anisotropic topological derivatives framework. In addition, the dense arrangement of fibers was evenly spaced for the part's manufacturability while retaining their specific patterns. However, the structural performance resulting from simplifying the dense fiber arrangement was not evaluated, thus questioning the reliability of the printed part.

The work, as mentioned earlier, implemented mono-scale approaches to optimize the distribution and orientation of the FRC material. However, CF4 also provides an effective way to fabricate mono-scale structures and multi-scale structures. Multi-scale approaches can be classified into lattice-based and hierarchical-based topology optimization based on their micro-scale optimization methodology. The first approach uses offline homogenization to calculate effective elastic coefficients of a fixed or a set of multi-variable micro-structures. The computed sets of effective elastic coefficients are then interpolated to attain continuous variations in these coefficients for the corresponding variations in the micro-structure's parameters. In contrast, the hierarchical approach solves a spatially varying micro-structural and macro-scale optimization problem at each material point, leading to high computational costs and connectivity issues. Nevertheless, since FRCs have inherent multi-scale characteristics, spatially varying material distributions and geometric patterns spanning at least two or more scales hold a promising future for designing next-generation lightweight structures. In addition, interested readers can refer to Wu et al. [129] review paper to understand the general framework for multi-scale topology optimization.

On the other hand, the multi-scale strategy for anisotropic materials is challenging for the following reasons: length scale controls, connectivity across adjacent micro-structures, ability to produce models for damage criteria to capture actual anisotropic behavior, and unique treatments at the boundaries of the domain, for example. One must investigate these factors through experiments or appropriate numerical tools to estimate the actual performance of printed parts. Only a few works address the multi-scale approach for FRCs based on the authors' knowledge. Kim et al. [130] adopted the homogenization method for simultaneously designing spatially varying fiber volume fractions and orientations as shown in Figure 6. Their work used SIMP to design the macro-structure's composite topology. Finally, the de-homogenization procedure [131] applied to fiber micro-structures obtained in the coarser mesh was visualized by projecting at a finer mesh. Various benchmark and multi-load structure problems have been studied, and it was concluded that locally varying FRC materials augment the global stiffness of the structure more than a fixed fiber volume fraction or isotropic multi-material structure. In continuation of the Kim methodology, Jung [132] proposed a 3D TO approach for designing FRC structures with spatially varying fiber fractions and orientations. Finally, a work by Boddeti et al. [133] introduced a complete design to the manufacturing workflow for laminated continuous fiber-reinforced composites with variable stiffness enabled by spatially varying micro-structures.

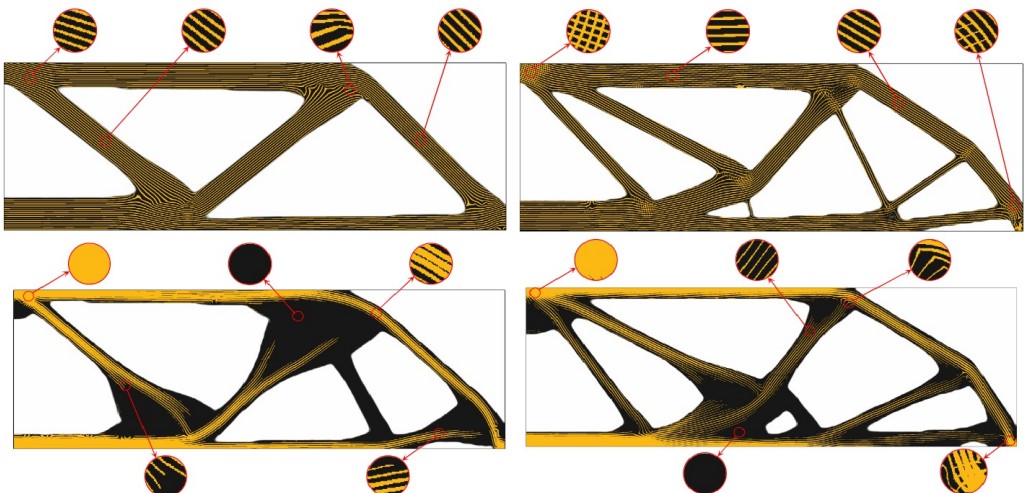

**Figure 6.** MBB—post-processed design result of FRC structure with fixed (**top-row**) and varying fiber volume fraction (**bottom-row**) using a micro-structure unit cell with rectangular-shape (**left-column**) and cross-shaped (**right-column**) fiber layout. The black color represents matrix material, and the yellow colored part is fiber material [130].

## 4. Discussion

The discussion focuses on the suitability of given topology optimization for anisotropic materials, given a pre-requisite understanding of the CF4 and its limitations. Therefore, the following discussion does not exhaustively address CF4 and its differences in adopting a particular TO method. Thus, the focus is on investigating the suitability of TO strategies that can fully exploit the design freedom offered by CF4 technologies. As mentioned earlier in Section 3, the existing techniques for material orientation are categorized into five major classes: continuous, discrete, discrete-continuous, feature, and material parameterization.

Optimizing a prescribed set of alternative discrete angles, referred to as the DMO, is often preferred in the aerospace, automotive, and wind turbine industries for manufacturability reasons. The DMO approach is favorable for multi-layer composite laminate designs [134–136] because a mixed-integer programming problem is formulated as a continuous problem that can be solved efficiently using gradient-based optimizers. As a result, substantial problems that might not be amenable to gradient-free methods can use DMO parameterization. Given the simplicity of DMO and the possibility of attaining the discrete setting for material orientation, it attracted various industrial applications with the flexibility to embrace different manufacturing units available in the market. However, restrictive DMO measures on material orientation design space may only partially exploit the potential of CF4, and the deduction is based on limited literature on its applicability to CF4.

Alternatively, optimization of the multi-layered composite laminates [137] can be attacked using an indirect approach, i.e., by applying lamination parameters, as introduced by Tsai and Pagano [138]. It parameterized composite laminate's stiffness utilizing twelve linearly interdependent parameters. Therefore, it is advantageous to reduce the number of optimization variables (independent of the number of layers) and introduce a convex design space, thus making it possible to obtain a globally optimal solution. However, lamination parameters (LPs) are not independent design variables; therefore, it is difficult to include, e.g., composite failure criteria and design and manufacturing constraints. In addition, it is limited to considering one candidate material and does not generate a direct description of the laminate data for the design. As a result, LPs require additional optimization steps to convert the stiffness properties to optimal fiber orientation angles to introduce design guidelines and manufacturing constraints. Therefore, a multi-level optimization strategy is used to exploit the benefits to achieve global optima at its first stage. However, despite these methods' popularity, it limits the capability of CF4 processes that enable multi-axial and

micro-scale prints. In addition, it must consider AM-related manufacturing constraints, e.g., minimum turning angles, feature sizes, etc., such that these constraints are concurrently evaluated in the optimization process to achieve realizable AM design. Thus, the framework currently does not offer scalability for numerical verification of the CF4 prints. The LPs framework is briefly discussed for completeness on available parameterization schemes. Still, interested readers can refer to the review by Albazzan et al. [139] and recent work on the TO of laminates in the following citations [140–143].

Continuous orientation methods naturally become a suitable parameterizing scheme for CF4 processes because these methods provide the highest freedom regarding shape and variable stiffness. Thus, the continuous orientation formulation directs the material deposition path planning to ensure the fiber trajectory curvature, fiber continuity, fiber fraction, and offset distance between adjacent fibers, unlike discrete methods, where fiber convergence and fiber continuity are challenging to attain. Papapetrou et al. [99] designed the topology and material orientation of parts simultaneously; the optimized results were post-processed using continuous fiber path planning to ensure realizability. A sequential scheme was proposed [13,144] where the fiber placement based on load transmission follows isotropic TO; this is contrary to Liu [145], who adopted concurrent fiber path planning and structural topology optimization. The multi-axis material deposition technology using a robotic arm requires an extension of the TO algorithm to envelop the 3D fiber orientation, in contrast to in-plane printing. Schmidt et al. [146] introduced azimuth and elevation angles to extend the CFO method for 3D fiber orientation. In addition, they emphasized the issues of the non-convexity of the compliance and sensitivity to the initial fiber orientations by investigating the orientation parameter space to mitigate these problems [147]. Finally, the realizability of 3D-printed composite is studied by Fedulov et al. [148], where they proposed a filtering technique for fast convergence.

Utilizing TO methods for exploring CF4 technology, generally speaking, heightens the composite manufacturing cost, especially when applying these technologies in the production of large-scale structure parts. Therefore, understanding the trade-off among commercial aspects, i.e., realizability, practicability, and structural design, requires assimilating the benefits of the discrete, continuous, and multi-component methodologies. Thus, a discrete-continuous parameterization scheme optimizes the structural topology and material orientation, including multi-component optimization (MTO) that decomposes product geometry while guaranteeing manufacturing constraints that might significantly impact the quality and cost of the end product. Initially, a genetic algorithm was used to solve MTO [149], and recently, a gradient-based optimization algorithm was used by Zhou et al. [150]. Zhou et al. [151] further extended their work for structures made of multiple composite components with tailored material orientations without a prescribed set of alternative discrete angles. Therefore, these methods can produce regions fabricated separately and joined with either continuous or discrete material orientation methods, or both, as considered in Qiu's work [91].

Feature-based parameterization follows the ideology of ready-for-manufacturability with a necessary restriction on the spatial distribution of the fiber orientations. It envelops commercial aspects for the realizability of composite parts by introducing CAD-based features to ease the manufacturing process with the potential for layerwise design. Moreover, it further simplifies the design space by considerably reducing design variables. It is noted that published works only considered stiffness-driven design. However, it is also critical to consider failure modes for composite parts manifested using the layerwise AM process. These failure criteria render markedly different designs that raise the method's relevance in fabricating FRC structures.

Figure 7 depicts optical microscopy images of the cross-section of the fiber-reinforced plastic material, namely, carbon fibers impregnated with nylon plastic (polyamide 6), which is wound on the filament spool placed in the material chamber, making a somewhat circular cross-section. However, the filament cross-section takes a rectangular shape once it is deposited on the platen. This is because the carbon fiber bundle filament passes through

the roller before it is deposited on the platen. In addition, He et al. [15] demonstrated the effects of voids on the failure mechanism due to poor fiber–matrix interfaces, causing reduced mechanical performance. Moreover, as high as 12% void content was observed for CF/PA6 composites, with a fiber volume fraction of 35%. Based on the experimental observation, naturally, the design space envelops the behavior of deposited material both at the micro- and macro-scale; thus, multi-scale models can predict the actual response of the print structure. Thus, multi-scale modeling with disposable parameterization schemes can fully exploit the capabilities of CF4. The benefits of the micro-mechanics approach are that it can predict all of the elastic properties and the composite's complex, multi-axial, nonlinear response based on the constituents' properties.

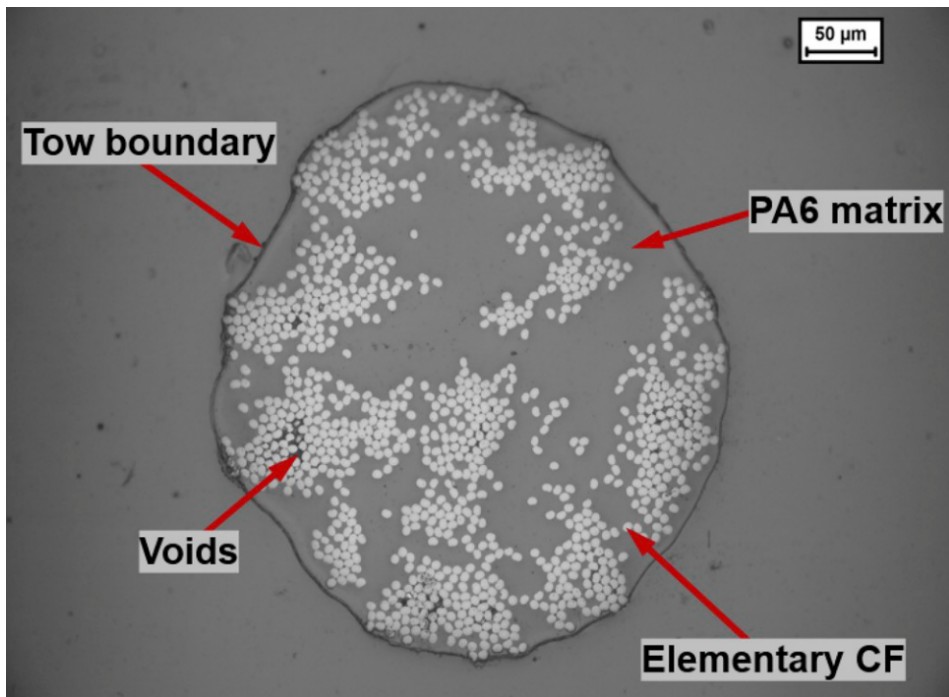

**Figure 7.** Typical cross-sectional view of a CF/PA6 filament by optical microscopy [15].

However, multi-scale modeling by no means eliminates the need for the mono-scale TO, but it can be considered an extension to the DfAM framework to understand the convoluted physics that bridges the materials and structures. Another outlook for DfAM research works might include multi-objective performance measures or extending their framework to include objectives other than compliance. Moreover, to enable manufacturing constraints, emerging feature-based topology optimization approaches allow for easy integration of such restrictions in the optimization process.

## 5. Conclusions

This work reviews TO methods for FRC structures applied to CF4. First, the study underlines the single-scale TO approaches that simultaneously or sequentially design fiber orientation and structural topologies. The study classified parameterizing techniques for anisotropic materials' topology optimization. Continuous parameterization schemes are considered for spatially varying fiber orientations and/or fiber volume fractions. In addition, these methods are amenable to CF4 and multi-axis material deposition technologies. Contrary to that, to attend to established manufacturable units, e.g., automatic fiber placement or automated tape layup, DMO is widely chosen to optimize composite laminates. The study further reports the usefulness of multi-scale TO for realizing FRC and extending it for variable fiber fraction structures. Moreover, it highlights emerging TO methodologies, such as feature-mapping, multi-component, and isogeometric optimization. The potential

applicability of these methods towards CF4 sets a new bar for designing FRC structures. Thus, we address the following main challenges in designing TO for anisotropic materials:

*General applicability*: Most studies utilize the performance measure to minimize compliance and material distribution on a simple structure; thus, these methodologies still require validation for complex problems, such as bucking stability, compliant mechanisms, eigenvalue analysis, etc.

*Solution dependency on an initial guess*: CFO methods are sensitive to starting guesses but are widely chosen for their simplicity and inherent attribute to the design of continuous fiber orientation. Taking the benefits of CFO and DMO provides a new direction to realize free-form FRC structures additively, but it still needs further improvement and research.

*Multiple constraints*: Simple volume constraints integrated with manufacturing constraints, such as minimum curvature, fiber filament cut-out, feature size, etc., are essential to validate the optimization process and realizability of the printed part.

*Revival of shape-based TO*: Several works adopt pixel- or voxel-based TO approaches due to their simple implementation but need not be a suitable choice to attain CAD-friendly design and not necessarily adapt to other FRC manufacturing units. Thus, emerging TO methodologies, such as feature mapping, isogeometric shape optimization, and multi-component methods, have been considered for the DfAM framework.

*Three-dimensional printing of continuous FRCs*: CF4 is a young technology but has favorable attitudes toward designing structures' mechanical properties, such as tensile, flexural, compression, and impact properties. However, the printed parts' strength will always be lower than traditional methods due to the low fiber content, poor interface bonding force, void formation, and inevitable printing limitations. Thus, to improve the performance of given CF4 prints, a parallel investigation of the material and morphological properties of continuous FRCs must go hand-in-hand with the TO framework.

Finally, the choice of the TO methods for designing complex FRC parts applied to CF4 requires various considerations; thus, this review highlights the different aspects of TO methods used for FRC structures to lay an essential foundation for researchers entering the field of the TO of additively manufactured continuous FRCs.

**Funding:** Financed by the European Union—NextGenerationEU (Grant n° CN_00000023). The opinions expressed are those of the authors only and should not be considered as representative of the European Union or the European Commission's official position. Neither the European Union nor the European Commission can be held responsible for them.

**Institutional Review Board Statement:** Not applicable.

**Informed Consent Statement:** Not applicable.

**Data Availability Statement:** Not applicable.

**Conflicts of Interest:** The authors declare no conflict of interest.

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
