# Peer review of "A Review on Topology Optimization Strategies for Additively Manufactured Continuous Fiber-Reinforced Composite Structures"

_applsci, doi:10.3390/app122111211_

Round 1

Reviewer 1 Report

The manuscript is well written. The reviewer would like to make some suggestions.

1. In discussions, the author must name all the classes.

2. The results must be precise. It seems that the discussion cannot be differentiated from conclusion. The authors must present only the findings and make their opinion according to the findings. The conclusion can be written in bullet points

Reviewer 2 Report

The current manuscript presents a review on topology optimization strategies for additively manufactured continuous  fiber-reinforced composite structure. The title is attractive and presenting detailed information. However, there are some issues to be considered as follows:

- The introduction section should include a critical review for the additively manufactured continuous fiber-reinforced composite structures according to the recent studies; including applied AM techniques, materials, characteristics, and limitations for fabrications.

- The subsection numbering needs to be revised.

- Some titles of sections and subsections should be adjusted such as section 3. 

-  It is recommended to present an applied example for each parameterization strategy from the literature.

- The conclusion section should be focused, using a bullet points style is recommended.   

Reviewer 3 Report

Remarks in comments for Editors.

I agree to show them to the Authors.

Round 2

Reviewer 1 Report

The manuscript may be accepted for publication as the authors have addressed all the review comments

Reviewer 2 Report

the revised manuscript is improved, most of the comments and recommendations are well addressed. However, there are some issues still need to be considered as follows:

- The manuscript text should be carefully check to avoid data redundancy.  

- The conclusion section should be focused and conceived. Some data and recommendations should be moved to the discussion section. 

Reviewer 3 Report

Dear Editor, Dear Authors,

I read the paper again and the reviews.

I still have little doubt that Applied Sciences is the best journal for this paper. But the article is generally good, so I support editing it without further corrections.
